# Restoration in Vertebral Compression Fractures (VCF): Effectiveness Evaluation Based on 3D Technology

**DOI:** 10.3390/jfb13020060

**Published:** 2022-05-17

**Authors:** David C. Noriega González, Francisco Ardura Aragón, Jesús Crespo Sanjuan, Silvia Santiago Maniega, Gregorio Labrador Hernández, María Bragado González, Daniel Pérez-Valdecantos, Alberto Caballero-García, Alfredo Córdova

**Affiliations:** 1Department of Surgery, Ophthalmology, Otorhinolaryngology and Physiotherapy, Faculty of Medicine, University of Valladolid, 47005 Valladolid, Spain; fardura@saludcastillayleon.es; 2Department of Orthopedic, Clinic University Hospital of Valladolid, 47005 Valladolid, Spain; jcrespos@saludcastillayleon.es (J.C.S.); ssantiagom@saludcastillayleon.es (S.S.M.); glabrador@saludcastillayleon.es (G.L.H.); mbragadog@saludcastillayleon.es (M.B.G.); 3Department of Biochemistry, Molecular Biology and Physiology, Health Sciences Faculty, GIR: “Physical Exercise and Aging”, University of Valladolid, Campus Universitario “Los Pajaritos”, 42004 Soria, Spain; danielperezvaldecantos@gmail.com (D.P.-V.); a.cordova@uva.es (A.C.); 4Department of Anatomy and Radiology, Health Sciences Faculty, GIR: “Physical Exercise and Aging”, University of Valladolid, Campus Universitario “Los Pajaritos”, 42004 Soria, Spain; alberto.caballero@uva.es

**Keywords:** vertebral fracture, anatomical restoration, intravertebral implant, height restoration, vertebral kyphosis reduction, 3D technology

## Abstract

There are few studies about anatomical reduction of the fractured vertebral body before stabilization for treatment of vertebral compression fracture (VCF). Although restoration on vertebral height has been useful, the reduction of fractured endplates is limited. The vertebra is part of a joint, and vertebral endplates must be treated like other weight-bearing joint to avoid complications. The aim of this study was to evaluate the feasibility of anatomic reduction of vertebral compression fracture, in different bone conditions, fracture types, and ages (VCF). Under methodological point of view, we followed different steps: first was the placement of two expandable titanium implants just below the fracture. Later, to push the fractured endplates into a more anatomical position, the implants were expanded. Finally, with the implants perfectly positioned, PMMA cement was injected to avoid any loss of correction. To evaluate the effectiveness of this procedure in anatomical fracture reduction, a method based on 3D CT reconstructions was developed. In this paper, we have developed the procedure in three case studies. In all of them, we were able to demonstrate the efficacy of this procedure to reduce the VCF. The percentage of correction of the kyphotic angle varied range between 49% and 62% with respect to the value after the fracture preoperative value. This was accompanied by a reduction of the pain level on the VAS scale around 50%. In conclusion, this novel approach to the vertebral fracture treatment (VCF) associated with 3D assessment have demonstrated the possibility of reducing the vertebral kyphosis angle and the vertebral endplate fractures. However, given the few cases presented, more studies are necessaries to confirm these results.

## 1. Introduction

In the ageing population, vertebral compression fractures (VCFs) are a real concern. Around 70,000 osteoporotic VCFs per year are reported in the United States [1]. The cost of osteoporotic fractures in 2005 was $17 billion, but the worst is that for 2025, the cost is expected to increase to 50% [2].

Usually, VCFs are presented with acute axial back pain, although sometimes it is chronic due to previous fractures. In this regard, Kim and Vaccaro [3] have observed in patients with severe osteoporosis that these fractures occur during the period of bed rest.

The symptoms experienced by patients suffering from VCFs alter patient’s quality of life. This is due to the functional disabilities that they present. In general, these patients present neurological deficits, chronic pain, and acute and progressive spinal deformity, which lead to mood changes [4,5,6,7].

Several authors [8,9,10] have demonstrated a close correlation between clinical problems and spinal deformity. Post-traumatic kyphosis is one of the most potentially serious deformities. In addition, as Musbahi et al. [11] have indicated, neurological problems appear, and therefore the key issue is how the neurological status is affected, if the spinal stability is compromised, and if the fracture is new or old [11].

Currently in the pain phase, the treatments can help. However, most important is that a complete technique can provide the anatomical restoration of the entire vertebral body, cortical ring, and endplates [12].

Now, two fundamental methods are being developed to reduce VCFs. One is indirect to reflect the effect on the vertebral body. For this, the ligamentotaxis effect is used through patient positioning. It is a conservative surgical treatment, such as vertebroplasty or percutaneous instrumentation. Another technique is to act directly on the vertebra itself, exerting direct forces on the bone, i.e., performing a balloon kyphoplasty. Using the dynamic mobility of the VCF, it is possible to evaluate if the effect of ligamentotaxis can effectively reduce the cortical ring, and this should have no effect on the central part of the endplate [13,14,15,16].

In 1987, Galibert and Deramond [17] first described vertebroplasty. The authors aimed to reduce the vertebral fracture by correct positioning of the patient. For this, they used the dynamic mobility of vertebral fracture. Faciszewski et al. [18] have acted in the same way, stabilising the fractured vertebra in situ. Later, percutaneous balloon kyphoplasty have permitted acting directly on the vertebral body using inflatable bone clamps.

However, in this procedure, the reduction obtained by the bone forceps during cement injection cannot be maintained after deflation [19]. Furthermore, the inflatable bone forceps tend to follow the path of least resistance, that do not necessarily coincide with the direction of reduction. Although those treatments are able to reduce the pain and improve quality of life [1,20,21,22,23,24,25,26], there is no evidence that endplate reduction occurs using only the effect of ligamentotaxis or balloon. Additionally, different patterns of disc injury and healing may be responsible for complications after incomplete treatment [27].

Wang et al. [28] have indicated that, in order to avoid the occurrence of late kyphosis, is necessary to reduce the vertebral kyphosis angle. Several authors [3,7,29] have reported that both the reduction of the vertebral kyphosis angle and the restoration of height are beneficial in pulmonary diseases, reducing the risk of mortality [30,31,32,33,34,35].

In view of the previously mentioned, anatomical restoration of the endplate should improve the results by decreasing pain resulting from poor biomechanical posture. Previously, we have reported about the application of 3D analysis in biomechanical studies in traumatic injuries on the spine from cadavers [36]. In this paper, we aim to test if a new surgical procedure can be effective in VCFs. We also aim to validate previous publications on clinical cases with different origins and bone quality in order to evaluate and program reconstructive surgery for vertebral fractures, independent of the fracture type, the patient’s age, and bone quality. We believe that with this method, based on a 3D system, for the approach to this type of fracture, by including a reduction of the cortical ring and an anatomical restoration of the endplate, it is possible to obtain an anatomical restoration of the fractured vertebra before vertebral stabilisation by injection of acrylic cement PMMA.

## 2. Materials and Methods

### 2.1. The Procedure

To carry out this study, we have developed a novel, minimally invasive procedure to achieve anatomical reduction of vertebral compression fractures. We have used innovative permanent, intravertebral, craniocaudal expandable titanium implants. Before the vertebral stabilization, we have used acrylic cement (PMMA) to have a more effective stabilization. The procedure is developed in two stages, one for reduction and one for stabilization. During the surgical intervention, fluoroscopic guidance was used to have an optimal control of all the process. The study was approved by the Ethics Committee CEIm Área de Salud Valladolid Este (Spain) (Ref. EPA-10-33).


First step:


Consistent in the reduction. In this phase, two intravertebral titanium implants were placed, by means of a transpedicular approach to the vertebral body, in closed position, expandable, permanent, and craniocaudal (5 mm diameter and 25 mm length). With the reduction, it is possible to have an optimal implant placement. The implants expand in only one direction. Based on the preoperative evaluations, and by controlling the placement of implants, the surgeon is able to reduce the fracture successfully. Later, and by means of the instrumentation, he can carry the implant placement to reduce the fractured vertebral body (anteroposterior placement, height in the vertebral body, and the angulation with respect to the broken plateau). After reduction of the fracture, by the expansions of the two implants, the next step is to stabilize the reduction and maintain the correction.


Second step:


Stabilization. Using the acrylic cement injected into the two implants, stabilization is obtained. During the process of implant expansion, trabecular bone windows are formed due to implant deployment. This will permit an ideal interdigitation of the cement, obtaining an optimal mechanical stabilization of the fracture reduction.

### 2.2. Evaluation of Anatomical Restoration

To know the vertebral height restoration in fractures after vertebroplasty, there exists no unanimous methodology. Moreover, existing methods are imprecise and can lead to great variability. Therefore, it is necessary to define a method that can be used to evaluate anatomical restoration. In our study, we have used 3D technology, that is definitive in this proposed new treatment [12,18,37]. The method has been developed by LBM ENSAM (Paris, France) to quantify the anatomical restoration. The exploration and interventions were carried in the Hospital Clínico Universitario de Valladolid. We have used a system: Revolution General Electric spectral CT, Healthcare. The protocol to be followed was based on the following parameters: tube voltage: between 80 and 140 kVp, milliamperage: 190 mA, rotation time: 0.8 s, pitch: 0.516, slice thickness: 0.625 mm, ASIR-V-40%, ASIR-V-40% and ASIR-V-40%. From millimetre axial slices of CT scans, 3D reconstructions were obtained using a segmentation technique (Figure 1 and Figure 2).

This method has been validated previously [38,39,40]. It permits comparing, in 3D, two vertebral reconstructions. Once the two 3D reconstructions were obtained for the same vertebra, before and after the intervention, they were superimposed (Figure 3). The posterior arch was affected by neither the VCF nor by the surgical procedure. The two 3D reconstructions are compared by calculation of the distance between the same point belonging to the vertebral body surface. These calculations can be presented by color-coded 3D mappings, in that the calculated distance is represented for any point with a specific colour depending on the measured value.

The precision of the measurements depends on the thickness of the tomography slices. In our application, each 3D reconstruction was obtained using millimetric slices, with a precision of ±1 mm. These 3D reconstructions also make possible to quantify the vertebral angular changes between the two endplates, without any bias. Therefore, based on the 3D reconstruction, the sagittal vertebral kyphosis angle was determined using the projection of the two vertebral endplates in the sagittal plane.

### 2.3. The Patients

In each patient, preoperative planning with both X-rays and CT scan was made. With this information, were evaluated the cortical ring, the pedicles, the posterior wall, and both endplates of the fractured vertebra. Based on this, the first task was to evaluate the stability of the involved vertebral body, and the second function was to define the operative strategy that should be used to reduce the vertebral fracture, with respect to the defects of the endplates.

### 2.4. Case 1

A 57-year-old female (she suffered an osteopenia condition previously controlled) worker in a restaurant. She presented a wedge fracture in L1, (Figure 4) after a fall due to unstable floor. To evaluate the pain, a visual analogue scale (VAS) was used. In this scale, she presents an 8 level. In addition, she showed a defect in the central part of the superior endplate, as was shown in the preoperative CT scan.

This woman was operated on the fifth day following the traumatic event, with two implants placed in L1, via a percutaneous bipedicular approach under general anesthesia. Additionally, in this circumstance, 2 mL of PMMA cement were injected in each implant to maintain the vertebra in its reduced position. The overall procedure duration was around 35 min.

### 2.5. Case 2

Male, 36 years old, without medical antecedents, who fell while climbing. He suffered an A3.1 fracture of L1, with loss of 35% of the vertebral body height. His pain was evaluated preoperatively using VAS, resulting in a 7.5 level. The superior endplate of vertebra suffered a global defect with an impact on the posterior wall, which was displaced toward the spinal canal. This man was operated 3 days after the event under general anesthesia. The procedure had a duration of 24 min. In the process, a total of 3.4 mL of mid viscosity cement was used. The patient was released home after 24 h.

### 2.6. Case 3

Male, 71-year-old, retired, affected by an osteoporotic bony condition. He experienced pain (9.1 on VAS scale) after a biconcave fracture in L1 (Figure 5). The imaging assessments showed that the two endplates have been affected by the fracture.

The patient was operated twenty-one days after his first symptoms, under general anesthesia. After the fracture reduction by means two intravertebral permanent implants, PMMA cement was injected in the vertebra, to maintain the correction. A total of 3 mL of PMMA were used in the left pedicle and 2.5 mL in the right pedicle. The surgeon took around 41 min to perform all the procedure.

## 3. Results

In all of each cases presented here (wedge, crush, and biconcave), the use of expandable intravertebral implants has shown that it is possible to reduce the fracture of the vertebral body. In addition, the anatomy of the vertebral endplates was restored before the stabilization with PMMA cement injection. None of the procedures were associated with postoperative complications. The pain control was effective in all cases, and none of the patients needed a postoperative bracing.

### 3.1. Case 1

In this case, the patient was discharged from hospital three days after surgery. He went back to work three months after surgery, with a reduction of pain around 64% (level 8 in preoperative with VAS, until 2.9 at hospital discharge). The procedure was developed without problems, except for a minor non-symptomatic leakage in the intervertebral disc. The traumatic kyphosis vertebral angle was reduced a 62% in the sagittal plane (from 9.7° preoperative to 3.7° at discharge). 3D reconstructions (Figure 6) show that the defects in the superior endplate were corrected (Figure 6). The 3D mapping allowed for a precise quantification of this restoration for the whole vertebral body, and especially for both endplates (Figure 7) as well as the cortical ring restoration evidenced on the X-ray examination. The procedure has allowed to restore the endplate defect increasing the superior 6–2.5 mm of height increase in 85% of the total surface of the superior endplate.

The patient’s follow-up at 3, 6, and 12 months showed the efficacy of the treatment. At 3 and 6 months, the pain (VAS scale) was reduced to 2, and to 1.5 at 12 months. With respect to the vertebral height maintenance angle, we have observed 3.7° at 3 months, 3.6° at 6 months, and 3.8° at 12 months.

### 3.2. Case 2

The patient was discharged 24 h after surgery. He went back to work and to sports 1 month later, without any pain. Traumatic kyphosis vertebral angles were reduced 55% (from 6.8° in preoperative to 3.1° at discharge) and 73% (from 9.20° in preoperative to 2.4° at discharge) in L1 and L2, respectively. The superior endplate restoration achieved was 4.4 mm, having an impact on the total surface of it. The procedure was performed using two implants in each vertebra. After 9 years of the surgical procedure, the vertebral correction remained intact, and the sagittal alignment was in normal parameters for his age and sex.

The follow-up of the patient at 3, 6, and 12 months shows the following data. At 3, 6, and 12 months, observed 0 in pain. With respect to the maintenance of vertebral height, we observed: at 3 months, L1 vertebral angle 3.1°, L2 vertebral angle 2.4°; at 6 months, L1 vertebral angle 3°, L2 vertebral angle 2.4°; at 12 months, L1 vertebral angle 3°, L2 vertebral angle 2.5°.

### 3.3. Case 3

The patient was hospital discharged five days after surgery with a significant reduction of pain (50%). The kyphosis vertebral angle was reduced 49% (from 13° in preoperative to 6.6° at discharge). With respect the anatomical reduction, the superior plateau was restored as shown in the 3D reconstruction. The 3D mapping shows that the two implants have allowed the reduction of fractured plateau previous the stabilisation with cement (Figure 8). Both superior and inferior endplates were reduced using the following procedure: between 2.5–3.5 mm of increase in height and 20% of the superior endplate, and between 1.5 mm to 2.5 mm of increase in height in 13% of the inferior endplate (Figure 8).

Follow-up of the patient at 3, 6, and 12 months showed the efficacy of the treatment. At 3 months, pain, in VAS, we observed 1.9, at 6 months 1.5 and 1.2 at 12 months. Regarding the vertebral angle, a measurement of 6.6° at 3 months, 6.8° at 6 months and 6.9° at 12 months.

## 4. Discussion

In this relevant study, based on the previous study [36] with cadaveric spines and normal spines, the surgical technique planned and developed has demonstrated the ability to anatomically reduce the broken fragments. This has led to an improvement in the functional status of the patients with a restoration of the biomechanical properties of the functional disc unit, with a clear impact on quality of life.

Traditionally, the surgical procedure for VCF was limited to aggressive open stabilization methods [41]. This procedure entailed multiple comorbidities. In addition, in some cases, the patient’s age was a limiting factor. More recently, for VCF, percutaneous treatments have been performed [42]. Galibert et al. [17] have introduced new factors when performing a percutaneous injection of viscous polymethylmethacrylate (PMMA) in the vertebral body for vertebroplasty. In the kyphoplasty process, a balloon is introduced percutaneously into the fractured vertebral body, which is inflated to create a cavity. Later, the balloon is deflated, removed, and PMMA is injected.

Now these techniques are widely used for the treatment of VCFs [43,44]. In the treatment of acute and painful VCFs, their efficacy has been demonstrated, with an increase on recovery time compared to conservative treatment [26]. However, despite the use of these techniques, we have not found evidence that anatomical reduction of VCFs involves restoration of height of vertebral body, reduction of vertebral kyphosis, and restoration of endplate morphology prior to stabilisation [1,22,45]. In addition, these two percutaneous techniques are developed, while injecting cement, to maintain reduction forces. In addition, in some cases, some loss of reduction occurs with balloon kyphoplasty [19].

Our results showed in this study demonstrate that anatomical vertebral restoration after VCF is possible if the device allows good control of the reductive actions, both in quantity and direction. While it is true that it is possible to reconstruct a depressed vertebral body in such a way that mobilization is not delayed, it is also true that this is an art that must be mastered and performed in a totally controlled manner. In the mid-1980s, in relation to the treatment of calcaneal fractures, Trickey [46] said that “there is nothing wrong with the idea; it is the execution that fails”.

In our work, we attempt to go beyond vertebral height restoration in the anatomic reduction of the vertebral body in the treatment of VCF. We think that this is key, because as another joint of the body, an adequate anatomical restoration leads to improve the functional and biomechanical functions.

Trickey [46] observed that injuries with a poor prognosis are those for which a joint surface is distorted with consequent impairment of movement and can be painful. Improved joint function can be achieved by means of a better remodelling of the joint surface.

Since the spine supports loads during everyday life and activities [47], vertebral joint injuries deserve to be treated on the same principles as any other injury of the weight-bearing joints—that is, by means of a biomechanical stable anatomical reduction to allow early mobilization, weight bearing, and biomechanical restoration.

From the clinical point of view, the restoration of vertebral height should be important [12]. In our opinion, in accordance with our previous data [36] and the implementation in terms of clinical and functional outcomes in relation to radiological findings, the restoration of the endplate should be even more important. Rohlmann [48] expressed himself in the same sense by considering that vertebral loads after kyphoplasty in osteoporotic patients are only evident if an almost total reduction of the fracture is achieved.

The change in the centre of gravity occurred after single or multiple fractures, is a problem that affects adjacent joints and gait. In this sense, Kuo et al. [49] have indicated that there is an adaptive mechanism. Therefore, the anatomical and biomechanical restoration of a fractured vertebra is possible with the benefit obtained to minimising compensatory mechanisms in gait.

In the treatment of vertebral fractures for the local symptoms, there are other therapeutic options. The field of physiotherapy is used in the treatment of low back pain. Although in patients with fragility fractures, it should be noted that it is not free of complications because it increases the risk of new fractures [50]. This is mainly due to the fact that this type of manipulative therapy can exceed the forces applied and pass the limits of resistance of an osteoporotic spine. Therefore, the restoration of the biomechanical balance and resistance of the different parts of the osteoporotic spine is necessary before applying any physiotherapeutic technique.

Finally, in the field of vertebral fractures, it would be desirable evolve to osseointegration and bone regeneration. This aspect has already been pointed out by Hsieh et al. [51] in work with animal models. Additionally, Codrea et al. [52] in his revision about biomaterials propose the use of CAP (calcium phosphate) cement for bone ingrowth, but clinical trials are still pending, or with addition of other biomaterials such as Mg, as proposed Bose et al. [53]. However, in humans, osseointegration intravertebral devices are still under study.

There are other options such as an anodised titanium implant, using a triboelectric nanogenerator (TENG), or a barium titanate with Ag to promote osteogenesis and prevent infections, which could be of interest to develop intravertebral implants and avoid the use of PMMA [54,55]. This type of coating represents a field of development to improve clinical and radiological results.

## 5. Limitations of This Study

The major limitation of the study is the small number of cases presented. This is conditioned by the special nature of the technique and the tools required. It is exceptional to have the possibility of using 3D in fracture restoration. In addition, other specialists are needed to collaborate in the development and application of the technique. In our case, we are fortunate to be able to carry out these interventions sporadically.

## 6. Conclusions

In our opinion, this new procedure applied brings new opportunities for the treatment of VCF. We have observed (in 3D) how the vertebral anatomy restoration before the stabilization is correlate with improved clinical outcomes.

In this study, we have gone further and used a 3D evaluation method to assess the effect of the procedure on vertebral levels. With this procedure, we have been demonstrating the feasibility of the anatomical restoration of the fractured vertebral body and its impact on the functional outcome of patients. All of this is regardless of age, fracture type, and bone quality. Undoubtedly, and in spite of there being few cases, we believe that the expectations of clinicians in terms of feasibility and control of the procedure, safety, and pain have been met.

## Figures and Tables

**Figure 1 jfb-13-00060-f001:**
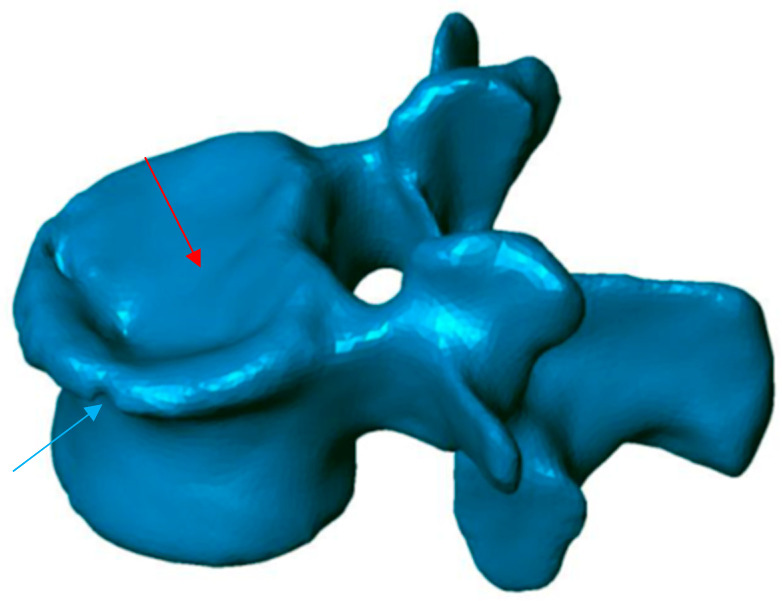
Preoperative 3D reconstruction. Sinking of the entire upper vertebral endplate (red arrow), with involvement of the cortical ring (blue arrow).

**Figure 2 jfb-13-00060-f002:**
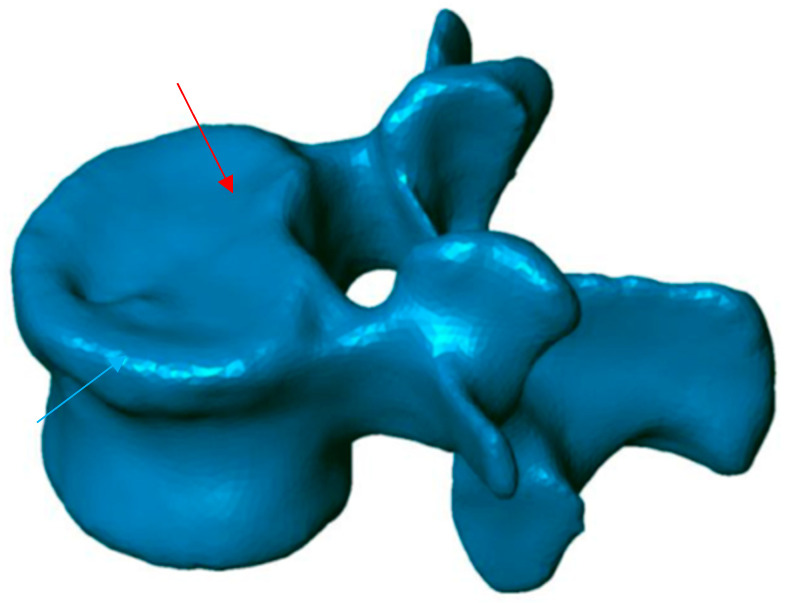
Postoperative 3D reconstruction. Reduction of most of the surface area of the upper end plate (red arrow) as well as the cortical ring (blue arrow).

**Figure 3 jfb-13-00060-f003:**
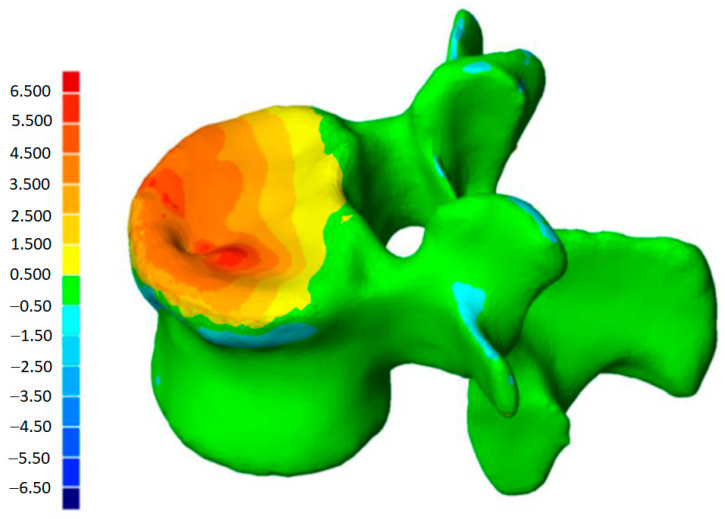
Color-coded 3D mapping. Superimposition of the preoperative and postoperative reconstructions, obtaining the altitude line, corresponding to the “mm” of reduction of the different parts of the vertebra.

**Figure 4 jfb-13-00060-f004:**
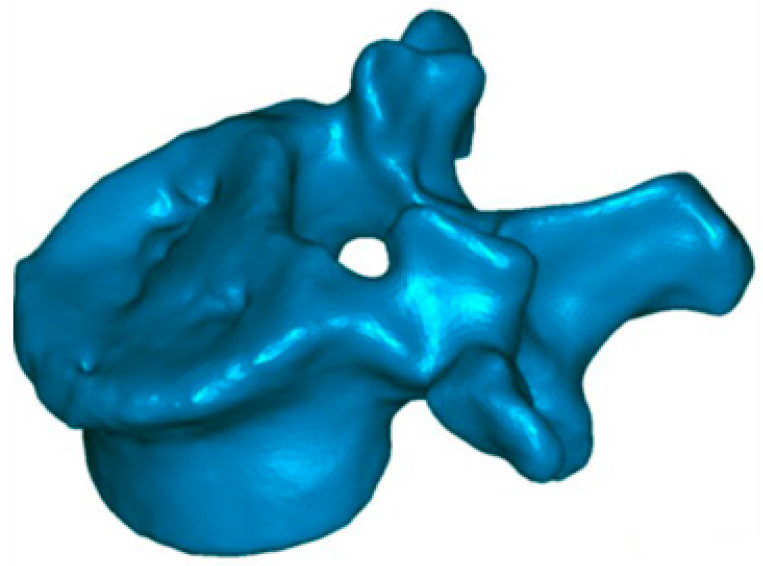
Central endplate defect. Preoperative reconstruction with extensive involvement of the upper vertebral plateau.

**Figure 5 jfb-13-00060-f005:**
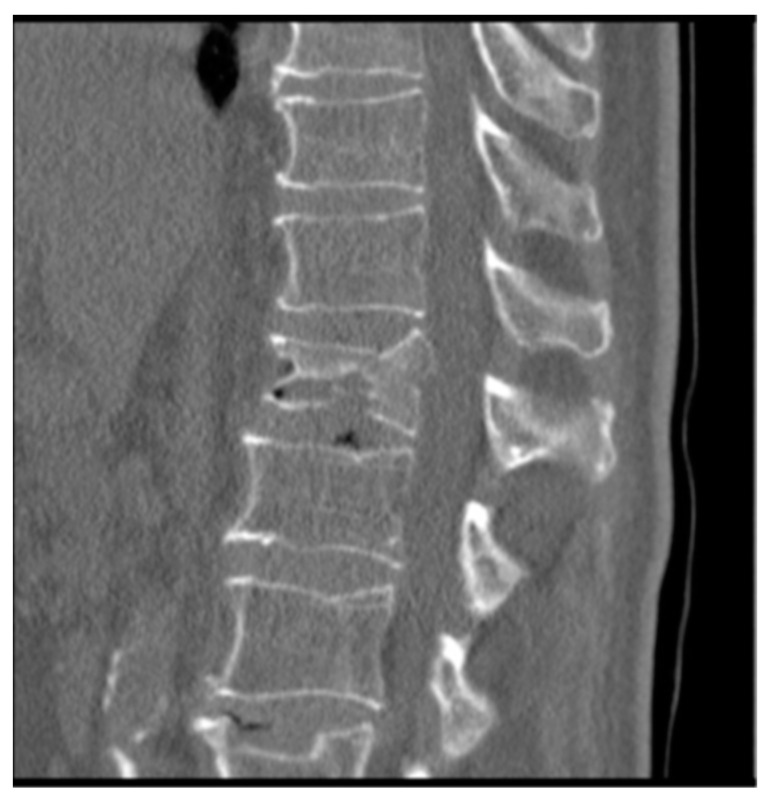
CT with both end plates impacted of the L1 vertebra.

**Figure 6 jfb-13-00060-f006:**
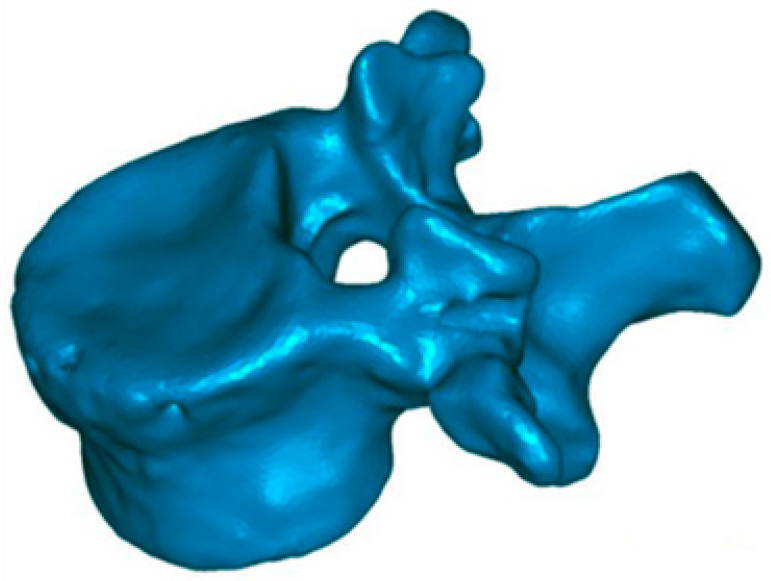
Superior endplate reduction. Preoperative reconstruction.

**Figure 7 jfb-13-00060-f007:**
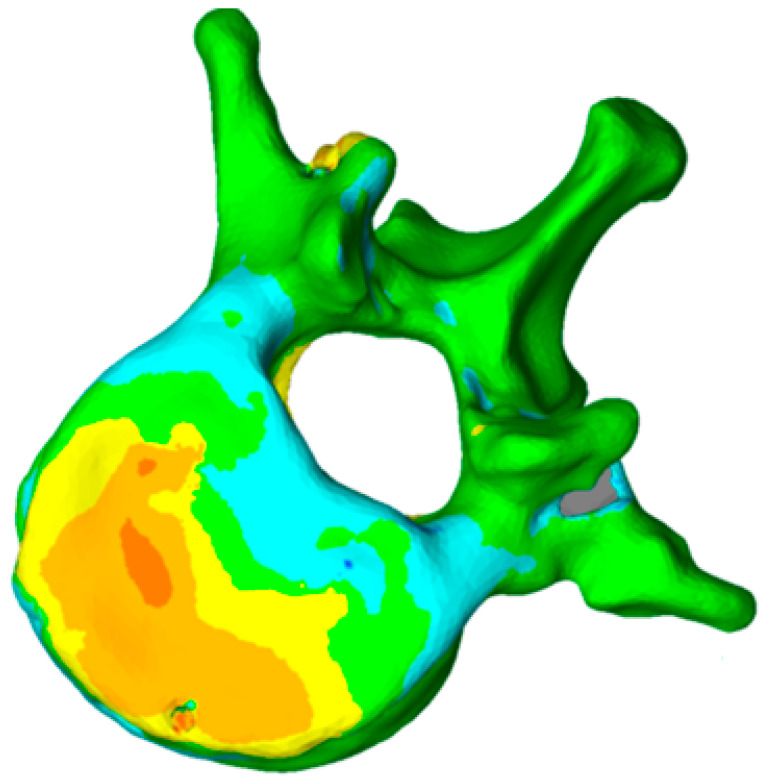
Topogram showing the 3D reduction of the vertebra superior endplate. The colour scale represents the reduction with red being the maximum and white the minimum (0–8 mm).

**Figure 8 jfb-13-00060-f008:**
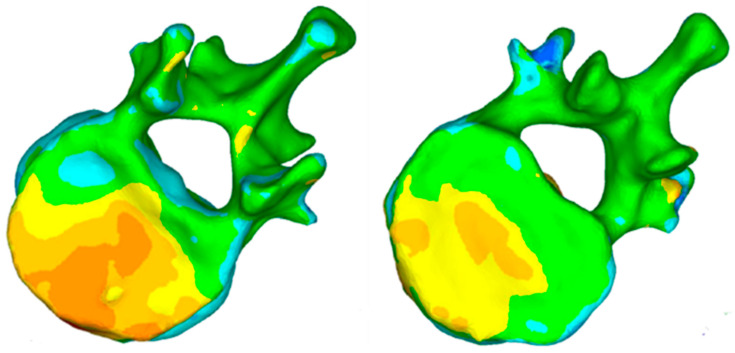
Topogram showing the 3D reconstruction of superior and inferior endplates. The colour scale represents the reduction with red being the maximum and white the minimum (0–8 mm).

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
