# Peer review of "Restoration in Vertebral Compression Fractures (VCF): Effectiveness Evaluation Based on 3D Technology"

_jfb, 2022, doi:10.3390/jfb13020060_

Round 1

Reviewer 1 Report

The purpose of this study was to evaluate the feasibility of anatomic reduction of vertebral compression fracture in different bone conditions, fracture type and ages. This study holds merit and the authors deserve recognition for their efforts in conducting this study. However, the authors should address some concerns before this article accepted for publication.

  1. The abstract lacks supporting evidence. The authors must provide sufficient quantitative data supporting their claims.
  2. Please provide more experimental information about the CT scanning, e.g. resolution, voltage, current in the section “2. Evaluation of anatomical restoration”.
  3. The authors must expand the “Discussion” section by providing a thorough interpretation of all their experimental results and include an exhaustive comparison between their experimental results and the exiting literature.
  4. Please add one more paragraph regarding the limitations of this study at the end of the “Discussion” section.
  5. Most of the references are over-dated. Please add more citation published in the recent 5 years. It’s very important to provide the latest research results, surgical or clinical cases for the reader and compared to the authors’ work.
  6. English editing is necessary.
  7. If possible, please consider to add the following reference in the manuscript:
  • TITLE: Posture Stability and Kinematics While Performing a 180° Turning Step in Elderly Individuals With and Without Vertebral Compression Fracture and in Middle-Aged Adults.
  • TITLE: Finite Element Investigation of Fracture Risk Under Postero-Anterior Mobilization on a Lumbar Bone in Elderly With and Without Osteoporosis.
  • TITLE: Comparison of Osseointegration in Different Intravertebral Fixators.

Author Response

REVIEWER 1.

The purpose of this study was to evaluate the feasibility of anatomic reduction of vertebral compression fracture in different bone conditions, fracture type and ages. This study holds merit and the authors deserve recognition for their efforts in conducting this study. However, the authors should address some concerns before this article accepted for publication.

  1. The abstract lacks supporting evidence. The authors must provide sufficient quantitative data supporting their claims.

The percentage of correction of the kyphotic angle varied between 49% and 62% with respect to the value after the fracture. This was accompanied by a reduction of the pain level on the VAS scale en torno a un 50%.

  1. Please provide more experimental information about the CT scanning, e.g. resolution, voltage, current in the section “2. Evaluation of anatomical restoration”.

The exploration and interventios were carried in the Hospital Clínico Universitario de Valladolid. We have uses a sistem: Revolution General Electric spectral CT, Healthcare. The protocol to be followed was based on the following parameters: tube voltage: between 80 and 140 kVp, milliamperage: 190 mA, rotation time: 0.8 sec, pitch: 0.516, slice thickness: 0.625 mm, ASIR-V-40%, ASIR-V-40% and ASIR-V-40%..

  1. The authors must expand the “Discussion” section by providing a thorough interpretation of all their experimental results and include an exhaustive comparison between their experimental results and the exiting literature.

We have expanded the discussion and introduced new bibliographical citations used in the discussion. All of these have been introduced in the body text of the manuscript.

  1. Please add one more paragraph regarding the limitations of this study at the end of the “Discussion” section.

Limitations of this study

The major limitation of the study is the small number of cases presented. This is conditioned by the special nature of the technique and the tools required. It is exceptional to have the possibility of using 3D in fracture restoration. In addition, other specialists are needed to collaborate in the development and application of the technique. In our case, we are fortunate to be able to carry out these interventions sporadically.

  1. Most of the references are over-dated. Please add more citation published in the recent 5 years. It’s very important to provide the latest research results, surgical or clinical cases for the reader and compared to the authors’ work.

The bibliographic citations suggested by you and others have been entered.

  1. English editing is necessary.

The text has been corrected by an expert

If possible, please consider to add the following reference in the manuscript:

We have introduced the 3 manuscripts that you have suggested. We have written 3 paragraphs, as discussion elements extracted from the suggested articles, and they have been placed at the end of the discussion, which we think has improved.

  1. Kuo, F.-C., Liao, Y.-Y., Lee, C.-H., Liau, B.-Y., & Pan, C.-C. (2020). Posture Stability and Kinematics While Performing a 180° Turning Step in Elderly Individuals With and Without Vertebral Compression Fracture and in Middle-Aged Adults. Journal of Medical and Biological Engineering, 40(2), 239–250. doi:10.1007/s40846-020-00508-9

The alteration of the centre of gravity occured after single or multiple fractures, is a problem that affects adjacent joints and gait. In this sense Kuo et al. [49], have indicated that there is an adaptive mechanism. Therefore, the anatomical and biomechanical restoration of a fractured vertebra is possible with the benefit obtained to minimising compensatory mechanisms in gait.

  1. Rungruangbaiyok, C., Azari, F., van Lenthe, G.H. Sloten JV, Tangtrakulwanich B, Chatpun S. Finite Element Investigation of Fracture Risk Under Postero-Anterior Mobilization on a Lumbar Bone in Elderly With and Without Osteoporosis. J. Med. Biol. Eng. 41, 285–294 (2021). https://doi.org/10.1007/s40846-021-00607-1

In the treatment of vertebral fractures for the local symptoms, there are other therapeutic options. The field of physiotherapy is used in the treatment of low back pain. Although in patients with fragility fractures it should be noted that it is not free of complications because increases the risk of new fractures [50]. This is mainly due because this type of manipulative therapy can exceed the forces applied and pass the limits of resistance of an osteoporotic spine. Therefore, the restoration of the biomechanical balance and resistance of the different parts of the osteoporotic spine is necessary before applying any physiotherapeutic technique.

  1. Hsieh, JY., Wang, JH., Chen, PQ. Huang YY. Comparison of Osseointegration in Different Intravertebral Fixators. J. Med. Biol. Eng. (2022). https://doi.org/10.1007/s40846-022-00698-4

Finally, in the field of vertebral fractures it would be desirable evolve to osseointegration and bone regeneration. This aspect has already been pointed out by Hsieh et al [51] in work with animal models. However, in human, osseointegration  intravertebral devices is still under study.

Reviewer 2 Report

Vertebral compression fractures (VCF) is a true concern in the aging population in the whole world. In this article, the aurhor used 3D analysis to evaluate the effect of the procedure on vertebral levels. 
However, this research is not innovative enough, the author should consider the following suggestions 
1. There are too many paragraphs in the introduction. The author can merge adjacent paragraphs appropriately 
2. The authors only evaluated the prognosis of patients in the short term. What about the prognosis in the 3 months, 6 months and one years?
3. The references are too old,  please add some references in recent years. It would be more valuable if the authors can present the latest studies about osteogenesis, referring to “Nano Energy, 2020, 74, 104825”, and “Chemical Engineering Journal, 2019, 374:304-315”.
4. Ther are some errors in the article, such as, line 25 "fracture type" should be corrected to "fracture types", line 104 "To carry out this study we have developed a novel" should be corrected to "To carry out this study, we have developed a novel".

Author Response

REVIEWER 2

Vertebral compression fractures (VCF) is a true concern in the aging population in the whole world. In this article, the aurhor used 3D analysis to evaluate the effect of the procedure on vertebral levels. 

However, this research is not innovative enough, the author should consider the following suggestions 

  1. There are too many paragraphs in the introduction. The author can merge adjacent paragraphs appropriately 

It has been corrected

  1. The authors only evaluated the prognosis of patients in the short term. What about the prognosis in the 3 months, 6 months and one years?

We have added more information for 1 year of follow-up.

Case 1

The patient's follow-up at 3, 6 and 12 months showed the efficacy of the treatment. At 3 and 6 months the pain (VAS scale) was reduced to 2, and to1.5 at 12 months. With respect the vertebral height maintenance angle, we have observed 3.7º at 3 months, 3.6º at 6 months and 3.8º at 12 months.

Case 2

The follow-up of the patient at 3, 6 and 12 months shown the follow data. At 3, 6 and 12 months observed 0 in pain. Respect the angle of maintenance of vertebral height, we observed: at 3 months, first angle 3.1°, second angle 2.4°; at 6 months, first angle 3°, second angle 2.4°; at 12 months, first angle 3°, second angle 2.5°.

Case 3

Follow-up of the patient at 3, 6 and 12 months showed the efficacy of the treatment. At 3 months, pain, in VAS, we observed 1.9, at 6 months 1.5 and 1.2 at 12 months. Regarding the vertebral height maintenance angle, we observed 6.6º at 3 months, 6.8º at 6 months and 6.9º at 12 months.

  1. The references are too old, please add some references in recent years. It would be more valuable if the authors can present the latest studies about osteogenesis, referring to “Nano Energy, 2020, 74, 104825”, and “Chemical Engineering Journal, 2019, 374:304-315”.

We have reviewed the documents you suggest and have expanded and introduced the comments in the discussion. We have introduced new bibliographic citations used for the discussion. All of these have been introduced in the body text of the manuscript.

  1. There are some errors in the article, such as, line 25 "fracture type" should be corrected to "fracture types", line 104 "To carry out this study we have developed a novel" should be corrected to "To carry out this study, we have developed a novel".

The errors have been corrected

Reviewer 3 Report

First of all the English needs to be improved. There are many sentences without sense which makes it difficult to understand the context of the manuscript. 

Although the topic is interesting it should be better presented. Especially the Figure legends have to be improved and the figures described in more detail. The references to the figures in the text are often missing. For exaple the differences between figure 1 and 2 should be pointed out by arrows, different colours, etc.

Author Response

REVIEWER 3

First of all, the English needs to be improved. There are many sentences without sense which makes it difficult to understand the context of the manuscript. 

Although the topic is interesting it should be better presented. Especially the Figure legends have to be improved and the figures described in more detail. The references to the figures in the text are often missing. For example, the differences between figure 1 and 2 should be pointed out by arrows, different colors, etc.

We have revised the text to indicate the tables. We have also expanded the figure legends.

Round 2

Reviewer 2 Report

This manuscript has been well revised.

Author Response

Thanks

Reviewer 3 Report

The authors have addressed all my comments and the manuscript improved. However, I do not believe that there was extensive English editing. just one example:

Usually VCFs are presented with acute axial back pain, although sometimes, is chronic by previous fractures. Even Kim and Vaccaro have indicate that these fractures, in patients with severe osteoporosis, occur during the patient is bedridden.

The first sentence is not a complete sentence and in the second there is a grammatical error. The manuscript should be edited by an english editing service.Then it can be accepted.

Author Response

The authors have addressed all my comments and the manuscript improved. However, I do not believe that there was extensive English editing. just one example:

We ask the publisher to correct the English and to invoice us for it

SENTENCES: Usually VCFs are presented with acute axial back pain, although sometimes, is chronic by previous fractures. Even Kim and Vaccaro have indicate that these fractures, in patients with severe osteoporosis, occur during the patient is bedridden.

The first sentence is not a complete sentence and in the second there is a grammatical error. The manuscript should be edited by an english editing service. Then it can be accepted.

SENTENCS CORRECTED:

Usually VCFs are presented with acute axial back pain, although sometimes, is chronic due to previous fractures. In this regard, Kim and Vaccaro [3] have observed in patients with severe osteoporosis that these fractures occur during the period of bed rest.